# Posterior Inferred, Now What?
# Streamlining Prediction in Bayesian Deep Learning

**Rui Li**    **Marcus Klasson**    **Arno Solin**    **Martin Trapp**
Department of Computer Science, Aalto University, Finland
`{firstname.lastname}@aalto.fi`

## Abstract

The rising interest in Bayesian deep learning (BDL) has led to a plethora of methods for estimating the posterior distribution. However, efficient computation of inferences, such as predictions, has been largely overlooked with Monte Carlo integration remaining the standard. In this work we examine streamlining prediction in BDL through a single forward pass without sampling. For this we use local linearisation on activation functions and local Gaussian approximations at linear layers. Thus allowing us to analytically compute an approximation to the posterior predictive distribution. We showcase our approach for both MLP and transformer architectures and assess its performance on regression and classification tasks.

See also extended paper at `https://arxiv.org/abs/2411.18425`.

## 1   Introduction

Through the success of machine learning models in real-world applications, ensuring their reliability and robustness has become a key concern. In particular, in applications such as aided medical diagnosis [1], autonomous driving [18], or supporting scientific discovery [22], providing reliable predictions, identifying failure modes, and identify how to reduce uncertainties of the system is vital. Uncertainty quantification is at the core of these topics with Bayesian deep learning (BDL, [27, 20]) providing a promising paradigm for assessing uncertainties effectively and efficiently.

The central goal in BDL is to make inferences w.r.t. the posterior distribution over the probabilistic model (the parameters or the function itself). For example, to compute the expected prediction, estimate model uncertainties, or use it within acquisition functions in active learning. For this, we need to first estimate the posterior distribution and secondly make inferences of interest based on the estimated posterior. While both of these steps typically involve intractable integration, only the first step has seen significant progress in recent years [3, 16, 4]. For the second step, in case of a Laplace approximation (LA, [11]), globally linearising the model function around the maximum *a posteriori* (MAP) estimate to perform inferences [13, 8] has shown promise in providing good predictive uncertainty. However, for all other posterior approximation methods, sampling based approximations remain to be the default. Given the high dimensionality of neural networks, sophisticated sampling methods are usually computationally prohibited and vanilla Monte-Carlo sampling is typically employed.

In this work, we tackle this problem by streamlining the prediction in BDL through local linearisation of activation functions and local Gaussian approximations at linear layers. Instead of a sample based approximation, which requires multiple re-evaluations of the network, we analytically approximate the posterior predictive distribution in a single forward pass through the network, making our methods well-suited for large-scale applications. Moreover, in contrast to global linearisation, our method is suitable for more complex inference tasks as the neural network function becomes locally linear with respect to the inputs. Empirically, we find that local linearisation and local Gaussian

Workshop on Bayesian Decision-making and Uncertainty, 38th Conference on Neural Information Processing Systems (NeurIPS 2024).

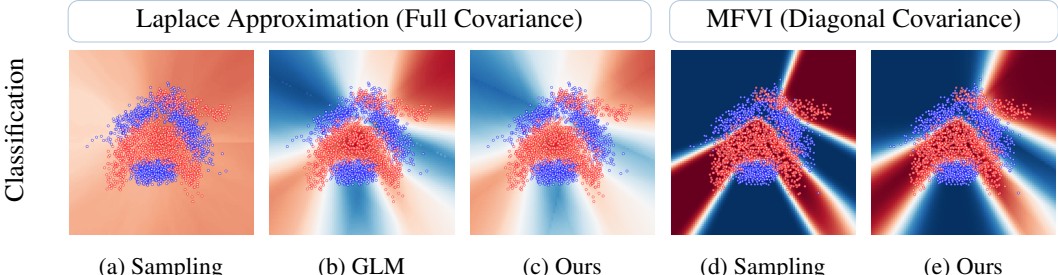

Figure 1: Ours gives better predictive uncertainties and decision boundaries compared with sampling in both Laplace approximation (LA) and mean-field variational inference (MFVI), while having matching performance with global linearised model (GLM) in LA.

approximation of neural networks to provide accurate predictive uncertainties and predictions, while being conceptually simple. Fig. 1 shows the posterior predictive densities for our proposal, compared to sampling based approximations and global linearisation in case of a Laplace approximation.

The contributions of our work can be summarised as follows: *(i)* We propose a sampling-free and deterministic method for approximating the posterior predictive distribution through local linearisation of activation functions and local Gaussian approximations in neural networks. *(ii)* We show how to exploit different covariance structures of the approximate posterior and present a streamlined prediction path for both MLP and transformer architectures. *(iii)* We evaluate our method on regression and classification tasks and find that our method result in good predictive performance.

## 2   Related Work

**Inferring Posterior in Bayesian Deep Learning**   There has been many methods developed which can be roughly grouped into three categories: *(i)* Laplace approximation based methods: Starting from [11] where simple post-hoc Laplace approximation (LA) has shown promising results, LA has gained increasing attention ever since. Recent works applied LA methods in various applications, such as large language models [29, 10] and dynamic neural networks [17]. *(ii)*: Variational inference (VI) based methods: [3] showed mean-field VI (MFVI) could improve generalisation in small-scale neural network and [24] showned MFVI is effective for large-scale neural networks as well. *(iii)*: Others: Monte Carlo Dropout [6] aims to estimate predictive uncertainty by interpreting dropout in neural networks as a form of Bayesian approximation. Deep ensemble [12] combines the outputs of multiple independently trained models to capture predictive uncertainty. Stochastic Weight Averaging-Gaussian [16], which extends Stochastic Weight Averaging [9] by capturing the posterior distribution of model weights using a Gaussian approximation.

**Making Prediction in Bayesian Deep Learning**   Little work has been done and the usual go-to solution is Monte Carlo Estimation. For Laplace approximation, [8] proposed the linearised LA by performing a global linearisation and has shown promise in providing useful predictive uncertainties.

## 3   Methods

In Bayesian deep learning (BDL), predicting the output $y$ (*e.g.*, class label, regression value) for an input $x \in \mathcal{X}$ is performed by *marginalizing* out the model parameters $\boldsymbol{\theta}$ of the neural network $f_{\boldsymbol{\theta}}(\cdot)$ instead of trusting a single point estimate, *i.e.*,

$$p(y \mid \boldsymbol{x}, \mathcal{D}) = \int_{\boldsymbol{\theta}} p(y \mid f_{\boldsymbol{\theta}}(\boldsymbol{x})) \, p(\boldsymbol{\theta} \mid \mathcal{D}) \, \mathrm{d}\boldsymbol{\theta}, \tag{1}$$

where $\mathcal{D} = \{(\boldsymbol{x}_n, y_n)\}_{n=1}^{N}$ denotes the training data and the posterior distribution $p(\boldsymbol{\theta} \mid \mathcal{D}) = \frac{p(\boldsymbol{\theta}, \mathcal{D})}{p(\mathcal{D})}$ is given by Bayes' rule. However, for most neural networks integrating over the high-dimensional parameter space is intractable, necessitating the use of approximations to compute the posterior distribution $p(\boldsymbol{\theta} \mid \mathcal{D})$ and the posterior predictive distribution $p(y \mid \boldsymbol{x}, \mathcal{D})$.

Let the weights and biases of the $m^{\text{th}}$ linear layer of the network $f$ be denoted as $\boldsymbol{W}^{(m)} \in \mathbb{R}^{D_{\text{out}} \times D_{\text{in}}}$ and $\boldsymbol{b}^{(m)} \in \mathbb{R}^{D_{\text{out}}}$, respectively. Then the pre-activation $\boldsymbol{h}^{(m)}$ is given as $\boldsymbol{h}^{(m)} = \boldsymbol{W}^{(m)} \boldsymbol{a}^{(m-1)} + \boldsymbol{b}^{(m)}$, where $\boldsymbol{a}^{(m-1)} \in \mathbb{R}^{D_{\text{in}}}$ is the activation of the previous layer. In case $m = 1$, then $\boldsymbol{a}^{(0)}$ corresponds to the input $\boldsymbol{x}$. We further denote the $k^{\text{th}}$ element of $\boldsymbol{h}^{(m)}$ as $h_k^{(m)} = \sum_{i=1}^{D_{\text{in}}} W_{ki}^{(m)} a_i^{(m-1)} + b_k^{(m)}$ and drop the superscript if it is clear from the context.

Given an approximate posterior distribution $q(\boldsymbol{\theta})$ with $\boldsymbol{\theta} = \{\boldsymbol{W}^{(m)}, \boldsymbol{b}^{(m)}\}_{m=1}^M$, we aim to compute the probability distribution of the activation $\boldsymbol{a}^{(m)}$ of each layer $m$. For this, we need to estimate the distribution of the pre-activation $\boldsymbol{h}^{(m)}$ and then compute an approximation to the activation $\boldsymbol{a}^{(m)}$ after application of a non-linear activation function $g(\cdot)$.

**Approximating the pre-activation distribution** In case the activation $\boldsymbol{a}^{(m-1)}$ is deterministically give, *i.e.*, for the input layer, we can compute the distribution over pre-activations analytically as a consequence of the stability of stable distributions under linear transformations [21]. However, for hidden layers the distribution over pre-activations is generally not of the same family as the posterior distribution [28]. Nevertheless, we will apply a local Gaussian approximation to the pre-activation at every hidden layer. Specifically, we make the assumption:

**Assumption 3.1.** *Assume that the activations of the previous layer $a_i^{(m-1)}$ and parameters of the $m^{th}$ layer are independent.*

Then followed by a Gaussian approximation of $a_i^{(m-1)} W_{ki}^{(m)}$ for each $i$ and each $k$, the mean of the pre-activation $\boldsymbol{h}^{(m)}$ is given as:

$$\mathbb{E}\left[\boldsymbol{h}^{(m)}\right] = \mathbb{E}\left[\boldsymbol{W}^{(m)}\right] \mathbb{E}\left[\boldsymbol{a}^{(m-1)}\right] + \mathbb{E}\left[\boldsymbol{b}^{(m)}\right], \tag{2}$$

and the covariance between the $k^{\text{th}}$ and the $j^{\text{th}}$ hidden unit is computed as:

$$\mathbb{C}\text{ov}\left[h_k^{(m)}, h_l^{(m)}\right] = \sum_{1 \le i,j \le D_{\text{in}}} \mathbb{C}\text{ov}\left[a_i^{(m-1)} W_{ki}^{(m)}, a_j^{(m-1)} W_{lj}^{(m)}\right] + \mathbb{C}\text{ov}\left[b_k^{(m)}, b_l^{(m)}\right]$$
$$+ \sum_{1 \le i \le D_{\text{in}}} \mathbb{E}\left[a_i^{(m-1)}\right] \left(\mathbb{C}\text{ov}\left[W_{ki}^{(m)}, b_l^{(m)}\right] + \mathbb{C}\text{ov}\left[W_{li}^{(m)}, b_k^{(m)}\right]\right), \tag{3}$$

where

$$\mathbb{C}\text{ov}\left[a_i^{(m-1)} W_{ki}^{(m)}, a_j^{(m-1)} W_{lj}^{(m)}\right] = \mathbb{E}\left[a_i^{(m-1)}\right] \mathbb{E}\left[a_j^{(m-1)}\right] \mathbb{C}\text{ov}\left[W_{ki}^{(m)}, W_{lj}^{(m)}\right]$$
$$+ \mathbb{E}\left[W_{ki}^{(m)}\right] \mathbb{E}\left[W_{lj}^{(m)}\right] \mathbb{C}\text{ov}\left[a_i^{(m-1)}, a_j^{(m-1)}\right]$$
$$+ \mathbb{C}\text{ov}\left[a_i^{(m-1)}, a_j^{(m-1)}\right] \mathbb{C}\text{ov}\left[W_{ki}^{(m)}, W_{lj}^{(m)}\right]. \tag{4}$$

Depending on the structure of the covariance matrix, we can further simplify the computation of the covariance matrix.

**Approximating the activation distribution** Let $g(\cdot)$ denote a non-linear activation function computing $\boldsymbol{a} = g(\boldsymbol{h})$ for a pre-activation $\boldsymbol{h}$. Inspired by the application of local linearisation in Bayesian filtering [*e.g.*, 23], we use a first order Taylor expansion of $g(\cdot)$ at the mean of the pre-activation $\mathbb{E}[\boldsymbol{h}]$. Specifically, we approximate $g(\boldsymbol{h})$ using

$$g(\boldsymbol{h}) \approx g(\mathbb{E}[\boldsymbol{h}]) + \boldsymbol{J}_g|_{\boldsymbol{h}=\mathbb{E}[\boldsymbol{h}]}(\boldsymbol{h} - \mathbb{E}[\boldsymbol{h}]), \tag{5}$$

where $\boldsymbol{J}_g|_{\boldsymbol{h}=\mathbb{E}[\boldsymbol{h}]}$ is the Jacobian of $g(\cdot)$ at $\boldsymbol{h} = \mathbb{E}[\boldsymbol{h}]$. Then, as stable distributions are closed under linear transformations, the distribution of $\boldsymbol{a}$ can be computed analytically and is given as follows in case of a Gaussian distributed, *i.e.*,

$$\boldsymbol{a} \sim \mathcal{N}(g(\mathbb{E}[\boldsymbol{h}]), \boldsymbol{J}_g|_{\boldsymbol{h}=\mathbb{E}[\boldsymbol{h}]}^\top \boldsymbol{\Sigma}_{\boldsymbol{h}} \boldsymbol{J}_g|_{\boldsymbol{h}=\mathbb{E}[\boldsymbol{h}]}). \tag{6}$$

Note that the quality of the local linearisation will depend on the scale of the distribution over the input $\boldsymbol{h}$. For ReLU activation functions, Petersen et al. [21] have shown that local linearisation provides the optimal Gaussian approximation of a univariate Gaussian distribution in total variation. For classification tasks, we employ a probit approximation [14, 11].

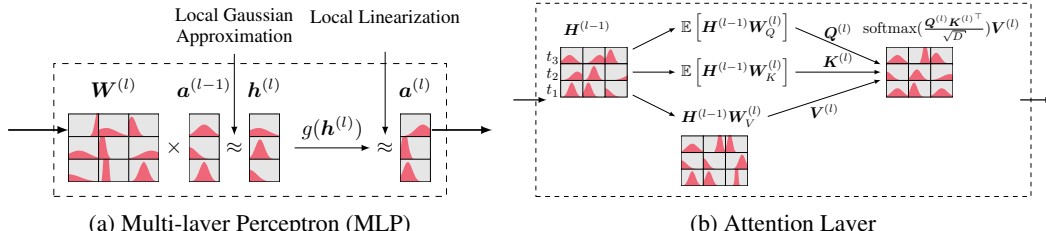

|                                    |                       |
|:----------------------------------:|:---------------------:|
| (a) Multi-layer Perceptron (MLP)   | (b) Attention Layer   |

Figure 2: Illustration our approach for different network architectures. In MLPs, we can directly apply local Gaussian approximations and local linearisation of each layer. The distribution over activations is then propagated to the next layer. In attention layers, we treat the query $Q$ and key $K$ deterministically and only treat the value $V$ as a random quantity, resulting in a straightforward propagation path. The resulting distribution is then propagated to the subsequent MLP layer.

Combining local Gaussian approximations for linear layers and local linearisation for non-linear activation functions results in a tractable approximation to the posterior predictive distribution, which can be computed in a single forward pass. Fig. 2 illustrates our streamlined prediction for multi-layer perceptrons (MLP) and attention blocks in tranformers, for a detailed description on the approach for transformers see App. B.6.

**Covariance Structure**   Computing the full covariance of the posterior is usually infeasible due to high computational and memory cost. Diagonal approximation and Kronecker-factorization of the covariance/precision are two of the most common aprpoaches. For diagonal covariance, calculating the posterior predictive distribution is straightforward, see App. B.2 for details. In case of Kronecker factors, we developed a tailored block retrieval method for efficient propagation of uncertainties, see App. B.3 for details. Note that other covariance structures can exploited in a similar fashion.

**Computational Complexity**   We will briefly discuss the computational complexity of our method for the case of full covariance. Observe from Eqs. (3) and (4) that the computational cost to obtain $(\mathbb{C}\mathrm{ov}[h_k, h_l])$ is $\mathcal{O}(\mathrm{D}_{\mathrm{in}}^{(l)}{}^2)$. Therefore, computing the output covariance at the $l^{\mathrm{th}}$ linear layer will be in the order of $\mathcal{O}(\mathrm{D}_{\mathrm{out}}^{(l)}{}^2 \mathrm{D}_{\mathrm{in}}^{(l)}{}^2)$. For element-wise activation functions, the computational cost will be $\mathcal{O}(\mathrm{D}_{\mathrm{out}}^{(l)}{}^2)$. Hence, we obtain a total cost of $\mathcal{O}(\sum_{l=1}^{L} \mathrm{D}_{\mathrm{out}}^{(l)}{}^2 \mathrm{D}_{\mathrm{in}}^{(l)}{}^2 + \mathrm{D}_{\mathrm{out}}^{(l)}{}^2)$ for a network with $L$ layers. By exploiting the covariance structure, the total computational cost can be substantially reduced.

## 4   Experiments

We adopt the Laplace approximation (LA, [15]) and mean-field variational inference [MFVI, 2] for approximating the posterior distribution of the network parameters. We compare our method using local Gaussian approximation and local linearisation against Monte Carlo (MC) sampling and a global linearised model [GLM, 8]. For MFVI, we adopt the IVON optimiser [24] to obtain the posterior approximation, which has been shown to be effective and scalable to large-scale classification tasks. Here, we compare our method against MC sampling from the posterior to make predictions as done in Shen et al. [24]. For the MFVI and LA sampling baselines, we used $1,000$ MC samples in the regression and MLP classification experiments, and $50$ MC samples for the ViT classification experiments. For our method, we addionally fit a scale factor, multiplied to the predictive variance, by minimizing the negative log predictive density (NLPD) on the validation set. This is necessary, as the predictive variance in case of deep and wide network with diagonal covariance structure can be large. We use a paired $t$-test with $p = 0.05$ and bold results with significant statistical difference.

**Regression**   We experiment with multi-layer perceptron (MLP) for regression. See App. C.1 for experiment setup details and additional results. We use full covariance for LA. As shown in Table Table 1, for MFVI our proposal (Ours) result in better performance than sampling on 8 data sets and matches the performance on the remaining 3 data sets. For LA, our approach obtains better performance than sampling on all data sets.

**Classification**   For MLP, we train it from scratch and treat all layers Bayesian. For ViT, we fine-tune the MLP layers in the last two blocks in a pre-trained Vision transformer (ViT) base model [5] and

Table 1: Negative log predictive density ↓ on UCI regression. Ours results in better or matching performance compared with sampling and GLM, indicating the effectiveness of our method.

| | $(n, d)$ | MFVI (Diagonal Covariance) | | Laplace Approximation (Full Covariance) | | |
| --- | --- | --- | --- | --- | --- | --- |
| | | Sampling | Ours | Sampling | GLM | Ours |
| SERVO | (167, 4) | $\mathbf{1.287}_{\pm0.069}$ | $\mathbf{1.136}_{\pm0.182}$ | $3.795_{\pm0.110}$ | $\mathbf{1.047}_{\pm0.172}$ | $1.443_{\pm0.077}$ |
| LD | (345, 5) | $\mathbf{1.346}_{\pm0.280}$ | $\mathbf{1.369}_{\pm0.440}$ | $2.221_{\pm0.110}$ | $\mathbf{1.495}_{\pm0.580}$ | $\mathbf{1.474}_{\pm0.648}$ |
| AM | (398, 7) | $1.004_{\pm0.052}$ | $\mathbf{0.807}_{\pm0.087}$ | $1.812_{\pm0.065}$ | $\mathbf{0.492}_{\pm0.279}$ | $\mathbf{0.478}_{\pm0.309}$ |
| REV | (414, 6) | $1.076_{\pm0.059}$ | $\mathbf{0.925}_{\pm0.091}$ | $1.932_{\pm0.045}$ | $0.859_{\pm0.129}$ | $\mathbf{0.833}_{\pm0.156}$ |
| FF | (517, 12) | $\mathbf{2.160}_{\pm3.003}$ | $\mathbf{2.333}_{\pm3.671}$ | $2.086_{\pm0.292}$ | $\mathbf{1.584}_{\pm0.950}$ | $\mathbf{1.596}_{\pm1.217}$ |
| ITT | (1020, 33) | $0.937_{\pm0.047}$ | $\mathbf{0.841}_{\pm0.065}$ | $1.681_{\pm0.069}$ | $0.825_{\pm0.095}$ | $\mathbf{0.756}_{\pm0.164}$ |
| CCS | (1030, 8) | $0.939_{\pm0.068}$ | $\mathbf{0.828}_{\pm0.108}$ | $1.612_{\pm0.048}$ | $0.319_{\pm0.109}$ | $\mathbf{0.234}_{\pm0.161}$ |
| ASN | (1503, 5) | $0.962_{\pm0.054}$ | $\mathbf{0.899}_{\pm0.065}$ | $1.788_{\pm0.045}$ | $0.422_{\pm0.109}$ | $\mathbf{0.396}_{\pm0.133}$ |
| CAC | (1994, 127) | $0.973_{\pm0.092}$ | $\mathbf{0.920}_{\pm0.118}$ | $1.848_{\pm0.055}$ | $\mathbf{1.281}_{\pm0.069}$ | $2.662_{\pm1.096}$ |
| PT | (5875, 19) | $0.976_{\pm0.069}$ | $\mathbf{0.940}_{\pm0.074}$ | $0.984_{\pm0.101}$ | $\mathbf{0.576}_{\pm0.181}$ | $0.651_{\pm0.306}$ |
| CCPP | (9568, 4) | $0.365_{\pm0.040}$ | $\mathbf{0.352}_{\pm0.042}$ | $1.345_{\pm0.085}$ | $\mathbf{-0.062}_{\pm0.182}$ | $\mathbf{-0.062}_{\pm0.200}$ |
| Bold Count | | 3/11 | 11/11 | 0/11 | 7/11 | 8/11 |

later treat them Bayesian. See App. C.2 for experiment setup details and additional results. With LA, we use a Kronecker-factorized covariance for MLPs and a diagonal covariance for ViT models. As shown in Table 2, for both MLP and ViT, we obtain better performance when compared with sampling and GLM.

Table 2: Negative log predictive density ↓ on classification data sets. Ours results in better or matching performance when compared with sampling, indicating the effectiveness of our approximation.

| | | MFVI (Diagonal Covariance) | | LA (Kron. Cov. for MLP, Diag. Cov. for ViT) | | |
| --- | --- | --- | --- | --- | --- | --- |
| | | Sampling | Ours | Sampling | GLM | Ours |
| MNIST | MLP | $0.179_{\pm0.014}$ | $\mathbf{0.086}_{\pm0.005}$ | $0.210_{\pm0.003}$ | $\mathbf{0.089}_{\pm0.004}$ | $\mathbf{0.089}_{\pm0.005}$ |
| FMNIST | MLP | $2.010_{\pm0.051}$ | $\mathbf{0.529}_{\pm0.011}$ | $0.556_{\pm0.008}$ | $0.548_{\pm0.018}$ | $\mathbf{0.397}_{\pm0.010}$ |
| CIFAR-10 | ViT | $0.124_{\pm0.011}$ | $\mathbf{0.080}_{\pm0.005}$ | $0.169_{\pm0.004}$ | $\mathbf{0.089}_{\pm0.005}$ | $\mathbf{0.088}_{\pm0.006}$ |
| CIFAR-100 | ViT | $0.480_{\pm0.018}$ | $\mathbf{0.437}_{\pm0.013}$ | $1.043_{\pm0.010}$ | $0.602_{\pm0.011}$ | $\mathbf{0.457}_{\pm0.012}$ |

**Robustness to Out-of-distribution** We now assess the robustness to out-of-distribution (OOD) data for our method and the baselines. In Fig. 3, we take the ViT network fine-tuned on CIFAR-10 and evaluate its predictive entropy on the SVHN data set [19]. Our method can distinguish between in-distribution and OOD data better than the LA MAP and MFVI Sampling. Although our method underfits on the in-distribution data, the separation between is clear for the OOD data similar. For results on MLP, see App. C.2.

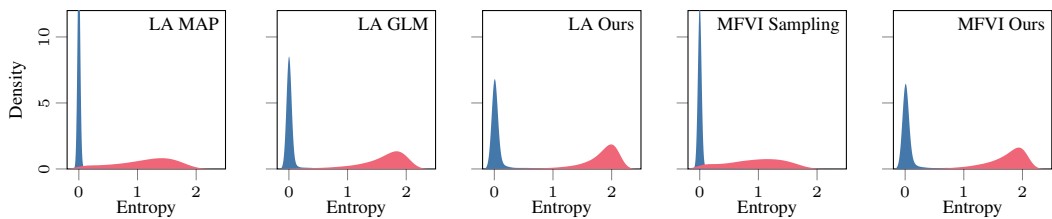

Figure 3: Kernel density plots over the predictive entropy from a ViT network finetuned on CIFAR-10 (blue, in-distribution) and data from SVHN (red, out-of-distribution). Our method results in a clear separation between the in- and out-of-distribution data.

## 5  Discussion & Conclusion

In this work, we proposed to streamline prediction in Bayesian deep learning by local linearisation and local Gaussian approximations. For this, we discussed the propgation in different neural network architecures and covariance structures. We showed through a series of experiments that our method obtains high predictive performance, obtain good predictive uncertainties, and can distinguish between in-distribution and OOD data. In future work, we aim to extend our approach to other network architectures, such as convolutional layers, and utilize our approach in more complex inference tasks.

## Acknowledgements

AS and RL acknowledge funding from the Research Council of Finland (grant number 339730). MT acknowledges funding from the Research Council of Finland (grant number 347279). MK acknowledges funding from the Finnish Center for Artificial Intelligence (FCAI). We acknowledge CSC – IT Center for Science, Finland, for awarding this project access to the LUMI supercomputer, owned by the EuroHPC Joint Undertaking, hosted by CSC (Finland) and the LUMI consortium through CSC. We acknowledge the computational resources provided by the Aalto Science-IT project.

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

# Posterior Inferred, Now What?
# Streamlining Prediction in Bayesian Deep Learning

## Supplementary Material

We first introduce notation in App. A. Then, we introduce the derivation of our method in App. B. At last, we describe the experiment setup and additional experiment results in App. C.

## A  Notation

We list notation that will be used throughout the appendix in Table 3.

Table 3: Notation.

| | |
|---|---|
| $\boldsymbol{x}$ | lowercase bolder letter, vector |
| $\boldsymbol{W}$ | uppercase bold letter, matrix |
| $\mathcal{D}$ | set |
| $x_i$ | $i^{\text{th}}$ element of $\boldsymbol{x}$ |
| $W_{ki}$ | $k^{\text{th}}$ row, $i^{\text{th}}$ column of $\boldsymbol{W}$ |
| $\boldsymbol{W}[k,:]$ | $k^{\text{th}}$ row of a matrix |
| $k, l$ | dimension of the output |
| $i, j$ | dimension of the input |
| $d$ | data feature dimension |
| $n, N$ | number of data points |
| $C$ | total number of classes |
| $m$ | layer index |

## B  Derivations

We derive the approximate posterior predictive distribution form in this section. App. B.1 is for the case where the covariance has full structure in linear layer. App. B.2 is for the case where the covariance has diagonal structure in linear layer. App. B.3 is for the case where the covariance has Kronecker-factorised structure in linear layer. App. B.4 is the derivation for activation layers. App. B.5 describes the probit approximation for approximate the posterior prediction for classification. App. B.6 describes how to apply our method for the transformer.

### B.1  Derivation for General Covariance Structure

Denote the weight and bias of the $m^{\text{th}}$ linear layer as $\boldsymbol{W}^{(m)} \in \mathbb{R}^{\text{D}_{\text{out}} \times \text{D}_{\text{in}}}$ and $\boldsymbol{b}^{(m)} \in \mathbb{R}^{\text{D}_{\text{out}}}$ respectively, and its input as $\boldsymbol{a}^{(m-1)} \in \mathbb{R}^{\text{D}_{\text{in}}}$. The pre-activation is then given as $\boldsymbol{h}^{(m)} = \boldsymbol{W}^{(m)}\boldsymbol{a}^{(m-1)} + \boldsymbol{b}^{(m)}$ with its $k^{\text{th}}$ element being $h_k^{(m)} = \sum_{i=1}^{\text{D}_{\text{in}}} W_{ki}^{(m)} a_i^{(m-1)} + b_k^{(m)}$.

We make the following assumptions to obtain a tractable distribution on the pre-activation:

- Assumption 1: We assume each $a_i^{(m-1)} W_{ki}^{(m)}$ is a Gaussian distribution.
- Assumption 2: We assume that the activations of the previous layer $a_i^{(m-1)}$ and parameters of the $m^{\text{th}}$ layer are independent.

From assumption 1, because now $a_i^{(m-1)} W_{ki}^{(m)}$ and $b_k^{(m)}$ are all Gaussian distributions, $h_k^{(m)}$ will follow Gaussian distribution as well. We call this local Gaussian approximation as we approximate each local component $a_i^{(m-1)} W_{ki}^{(m)}$ with a Gaussian. As now each $h_k^{(m)}$ is a Gaussian, $\boldsymbol{h}^{(m)}$ will be jointly Gaussian. We derive its mean and covariance and drop the layer index if it is clear from the context.

**Derivation of mean**  As $a_i$ is assumed to be uncorrected with $W_{ki}$, we have

$$\mathbb{E}[h_k] = \mathbb{E}\left[\sum_{i=1}^{D_{in}} W_{ki} a_i + b_k\right] \tag{7}$$

$$= \sum_{i=1}^{D_{in}} \mathbb{E}[W_{ki} a_i + b_k] \tag{8}$$

$$= \sum_{i=1}^{D_{in}} \mathbb{E}[W_{ki} a_i] + \mathbb{E}[b_k] \tag{9}$$

$$\approx \sum_{i=1}^{D_{in}} \mathbb{E}[W_{ki}] \mathbb{E}[a_i] + \mathbb{E}[b_k]. \tag{Assumption 2}$$

**Derivation of covariance**  The covariance between the $k^{\text{th}}$ and $l^{\text{th}}$ pre-activation can be written as

$$\mathbb{C}\text{ov}[h_k, h_l] = \mathbb{C}\text{ov}\left[\sum_{i=1}^{D_{in}} a_i W_{ki} + b_k, \sum_{i=1}^{D_{in}} a_i W_{li} + b_l\right] \tag{10}$$

$$= \mathbb{C}\text{ov}\left[\sum_{i=1}^{D_{in}} a_i W_{ki}, \sum_{i=1}^{D_{in}} a_i W_{li}\right] + \mathbb{C}\text{ov}\left[\sum_{i=1}^{D_{in}} a_i W_{ki}, b_l\right] + \mathbb{C}\text{ov}\left[\sum_{i=1}^{D_{in}} a_i W_{li}, b_k\right]$$
$$+ \mathbb{C}\text{ov}[b_k, b_l] \tag{11}$$

$$= \sum_{1 \leq i,j \leq D_{in}} \mathbb{C}\text{ov}[a_i W_{ki}, a_j W_{lj}] + \sum_{1 \leq i \leq D_{in}} (\mathbb{C}\text{ov}[a_i W_{ki}, b_l] + \mathbb{C}\text{ov}[a_i W_{li}, b_k])$$
$$+ \mathbb{C}\text{ov}[b_k, b_l] \tag{12}$$

We first derive the form of $\mathbb{C}\text{ov}[a_i W_{ki}, a_i W_{li}]$:

$$\mathbb{C}\text{ov}[a_i W_{ki}, a_j W_{lj}]$$

$$= \mathbb{E}[(a_i W_{ki} - \mathbb{E}[a_i W_{ki}])(a_j W_{lj} - \mathbb{E}[a_j W_{lj}])] \tag{13}$$

$$= \mathbb{E}[a_i W_{ki} a_j W_{lj} - a_i W_{ki} \mathbb{E}[a_j W_{lj}] - \mathbb{E}[a_i W_{ki}] a_j W_{lj} + \mathbb{E}[a_i W_{ki}] \mathbb{E}[a_j W_{lj}]] \tag{14}$$

$$= \mathbb{E}[a_i a_j W_{ki} W_{lj}] - \mathbb{E}[a_i W_{ki}] \mathbb{E}[a_j W_{lj}] - \mathbb{E}[a_i W_{ki}] \mathbb{E}[a_j W_{lj}] + \mathbb{E}[a_i W_{ki}] \mathbb{E}[a_j W_{lj}] \tag{15}$$

$$\approx \mathbb{E}[a_i a_j] \mathbb{E}[W_{ki} W_{lj}] - \mathbb{E}[a_i] \mathbb{E}[W_{ki}] \mathbb{E}[a_j] \mathbb{E}[W_{lj}] \tag{Assumption 2}$$

$$= (\mathbb{E}[a_i] \mathbb{E}[a_j] + \mathbb{C}\text{ov}[a_i, a_j])(\mathbb{E}[W_{ki}] \mathbb{E}[W_{lj}] + \mathbb{C}\text{ov}[W_{ki}, W_{lj}])$$

$$\qquad - \mathbb{E}[a_i] \mathbb{E}[W_{ki}] \mathbb{E}[a_j] \mathbb{E}[W_{lj}] \tag{16}$$

$$= \mathbb{E}[a_i] \mathbb{E}[a_j] \mathbb{C}\text{ov}[W_{ki}, W_{lj}] + \mathbb{E}[W_{ki}] \mathbb{E}[W_{lj}] \mathbb{C}\text{ov}[a_i, a_j] + \mathbb{C}\text{ov}[a_i, a_j] \mathbb{C}\text{ov}[W_{ki}, W_{lj}]. \tag{17}$$

Then we drive the form of $\mathbb{Cov}[a_i W_{ki}, b_l]$:

$$\mathbb{Cov}\left[a_i W_{ki}, b_l\right] = \mathbb{E}\left[(a_i W_{ki} - \mathbb{E}\left[a_i W_{ki}\right])(b_l - \mathbb{E}\left[b_l\right])\right] \tag{18}$$

$$\approx \mathbb{E}\left[(a_i W_{ki} - \mathbb{E}\left[a_i\right]\mathbb{E}\left[W_{ki}\right])(b_l - \mathbb{E}\left[b_l\right])\right] \tag{Assumption 2}$$

$$= \mathbb{E}\left[a_i W_{ki} b_l - a_i W_{ki}\mathbb{E}\left[b_l\right] - \mathbb{E}\left[a_i\right]\mathbb{E}\left[W_{ki}\right]b_l + \mathbb{E}\left[a_i\right]\mathbb{E}\left[W_{ki}\right]\mathbb{E}\left[b_l\right]\right] \tag{19}$$

$$= \mathbb{E}\left[a_i W_{ki} b_l\right] - \mathbb{E}\left[a_i\right]\mathbb{E}\left[W_{ki}\right]\mathbb{E}\left[b_l\right] \tag{20}$$

$$\approx \mathbb{E}\left[a_i\right]\mathbb{E}\left[W_{ki} b_l\right] - \mathbb{E}\left[a_i\right]\mathbb{E}\left[W_{ki}\right]\mathbb{E}\left[b_l\right] \tag{Assumption 2}$$

$$= \mathbb{E}\left[a_i\right]\left(\mathbb{E}\left[W_{ki}\right]\mathbb{E}\left[b_l\right] + \mathbb{Cov}\left[W_{ki}, b_l\right]\right) - \mathbb{E}\left[a_i\right]\mathbb{E}\left[W_{ki}\right]\mathbb{E}\left[b_l\right] \tag{21}$$

$$= \mathbb{E}\left[a_i\right]\mathbb{Cov}\left[W_{ki}, b_l\right]. \tag{22}$$

Putting it together, we have $\mathbb{Cov}[h_k, h_l] =$

$$\sum_{1 \le i,j \le \mathrm{D_{in}}} \mathbb{Cov}\left[a_i W_{ki}, a_j W_{lj}\right] + \sum_{i=1}^{\mathrm{D_{in}}} \left(\mathbb{E}\left[a_i\right]\mathbb{Cov}\left[W_{ki}, b_l\right] + \mathbb{E}\left[a_i\right]\mathbb{Cov}\left[W_{li}, b_k\right]\right) + \mathbb{Cov}\left[b_k, b_l\right], \tag{23}$$

where $\mathbb{Cov}[a_i W_{ki}, a_j W_{lj}] =$

$$\mathbb{E}\left[a_i\right]\mathbb{E}\left[a_j\right]\mathbb{Cov}\left[W_{ki}, W_{lj}\right] + \mathbb{E}\left[W_{ki}\right]\mathbb{E}\left[W_{lj}\right]\mathbb{Cov}\left[a_i, a_j\right] + \mathbb{Cov}\left[a_i, a_j\right]\mathbb{Cov}\left[W_{ki}, W_{lj}\right]. \tag{24}$$

Note that $\sum_{1 \le i,j \le \mathrm{D_{in}}} \mathbb{Cov}[a_i W_{ki}, a_j W_{lj}]$ in Eq. (23) could be rewrite into the form of matrix multiplication for efficient implementation:

$$\sum_{1 \le i,j \le \mathrm{D_{in}}} \mathbb{Cov}\left[a_i W_{ki}, a_j W_{lj}\right] \tag{25}$$

$$= \sum_{1 \le i,j \le \mathrm{D_{in}}} \mathbb{E}\left[a_i\right]\mathbb{E}\left[a_j\right]\mathbb{Cov}\left[W_{ki}, W_{lj}\right] + \mathbb{E}\left[W_{ki}\right]\mathbb{E}\left[W_{lj}\right]\mathbb{Cov}\left[a_i, a_j\right] + \mathbb{Cov}\left[a_i, a_j\right]\mathbb{Cov}\left[W_{ki}, W_{lj}\right] \tag{26}$$

$$= \sum \begin{bmatrix} \mathbb{E}\left[a_1\right]\mathbb{E}\left[a_1\right]\mathbb{Cov}[W_{k1}, W_{l1}] & \cdots & \mathbb{E}\left[a_1\right]\mathbb{E}\left[a_{\mathrm{D_{in}}}\right]\mathbb{Cov}[W_{k1}, W_{l\,\mathrm{D_{in}}}] \\ \vdots & \vdots & \vdots \\ \mathbb{E}\left[a_{\mathrm{D_{in}}}\right]\mathbb{E}\left[a_1\right]\mathbb{Cov}[W_{k\,\mathrm{D_{in}}}, W_{l1}] & \cdots & \mathbb{E}\left[a_1\right]\mathbb{E}\left[a_{\mathrm{D_{in}}}\right]\mathbb{Cov}[W_{k\,\mathrm{D_{in}}}, W_{l\,\mathrm{D_{in}}}] \end{bmatrix} \tag{27}$$

$$\odot \begin{bmatrix} \mathbb{Cov}[W_{k1}, W_{l1}] & \cdots & \mathbb{Cov}[W_{k1}, W_{l\,\mathrm{D_{in}}}] \\ \vdots & \vdots & \vdots \\ \mathbb{Cov}[W_{k\,\mathrm{D_{in}}}, W_{l1}] & \cdots & \mathbb{Cov}[W_{k\,\mathrm{D_{in}}}, W_{l\,\mathrm{D_{in}}}] \end{bmatrix} \tag{28}$$

$$+ \sum \begin{bmatrix} \mathbb{E}\left[W_{k1}\right]\mathbb{E}\left[W_{l1}\right] & \cdots & \mathbb{E}\left[W_{k1}\right]\mathbb{E}\left[W_{l\,\mathrm{D_{in}}}\right] \\ \vdots & \vdots & \vdots \\ \mathbb{E}\left[W_{k\,\mathrm{D_{in}}}\right]\mathbb{E}\left[W_{l1}\right] & \cdots & \mathbb{E}\left[W_{k\,\mathrm{D_{in}}}\right]\mathbb{E}\left[W_{l\,\mathrm{D_{in}}}\right] \end{bmatrix} \odot \begin{bmatrix} \mathbb{Cov}[a_1, a_1] & \cdots & \mathbb{Cov}[a_1, a_{\mathrm{D_{in}}}] \\ \vdots & \vdots & \vdots \\ \mathbb{Cov}[a_{\mathrm{D_{in}}}, a_1] & \cdots & \mathbb{Cov}[a_{\mathrm{D_{in}}}, a_{\mathrm{D_{in}}}] \end{bmatrix} \tag{29}$$

$$+ \sum \begin{bmatrix} \mathbb{Cov}[a_1, a_1] & \cdots & \mathbb{Cov}[a_1, a_{\mathrm{D_{in}}}] \\ \vdots & \vdots & \vdots \\ \mathbb{Cov}[a_{\mathrm{D_{in}}}, a_1] & \cdots & \mathbb{Cov}[a_{\mathrm{D_{in}}}, a_{\mathrm{D_{in}}}] \end{bmatrix} \odot \begin{bmatrix} \mathbb{Cov}[W_{k1}, W_{l1}] & \cdots & \mathbb{Cov}[W_{k1}, W_{l\,\mathrm{D_{in}}}] \\ \vdots & \vdots & \vdots \\ \mathbb{Cov}[W_{k\,\mathrm{D_{in}}}, W_{l1}] & \cdots & \mathbb{Cov}[W_{k\,\mathrm{D_{in}}}, W_{l\,\mathrm{D_{in}}}] \end{bmatrix} \tag{30}$$

### B.2 Derivation for Diagonal Covariance Structure

When the posterior has diagonal covariance, the mean $\mathbb{E}[h_k]$ will still be the same.

For covariance, note that as now the posterior is diagonal, when $k \neq l$, we have $\mathbb{Cov}[h_k, h_l] =$

$$\sum_{1 \leq i,j \leq \mathrm{D_{in}}} \mathbb{Cov}\left[a_i W_{ki}, a_j W_{lj}\right] + \sum_{i=1}^{\mathrm{D_{in}}} \left(\mathbb{E}\left[a_i\right] \mathbb{Cov}\left[W_{ki}, b_l\right] + \mathbb{E}\left[a_i\right] \mathbb{Cov}\left[W_{li}, b_k\right]\right) + \mathbb{Cov}\left[b_k, b_l\right] \tag{31}$$

$$= \sum_{1 \leq i,j \leq \mathrm{D_{in}}} \mathbb{Cov}\left[a_i W_{ki}, a_j W_{lj}\right] \tag{32}$$

$$= \sum_{1 \leq i,j \leq \mathrm{D_{in}}} \mathbb{E}\left[a_i\right] \mathbb{E}\left[a_j\right] \mathbb{Cov}\left[W_{ki}, W_{lj}\right] + \mathbb{E}\left[W_{ki}\right] \mathbb{E}\left[W_{lj}\right] \mathbb{Cov}\left[a_i, a_j\right] + \mathbb{Cov}\left[a_i, a_j\right] \mathbb{Cov}\left[W_{ki}, W_{lj}\right] \tag{33}$$

$$= \sum_{1 \leq i,j \leq \mathrm{D_{in}}} \mathbb{E}\left[W_{ki}\right] \mathbb{E}\left[W_{lj}\right] \mathbb{Cov}\left[a_i, a_j\right] \tag{34}$$

For $k = l$, we have $\mathbb{Var}[h_k] =$

$$\sum_{1 \leq i,j \leq \mathrm{D_{in}}} \mathbb{Cov}\left[a_i W_{ki}, a_j W_{kj}\right] + \sum_{i=1}^{\mathrm{D_{in}}} \left(\mathbb{E}\left[a_i\right] \mathbb{Cov}\left[W_{ki}, b_k\right] + \mathbb{E}\left[a_i\right] \mathbb{Cov}\left[W_{ki}, b_k\right]\right) + \mathbb{Var}\left[b_k\right] \tag{35}$$

$$= \sum_{1 \leq i \leq \mathrm{D_{in}}} \mathbb{Cov}\left[a_i W_{ki}, a_i W_{ki}\right] + \mathbb{Var}\left[b_k\right] \tag{36}$$

$$= \sum_{1 \leq i \leq \mathrm{D_{in}}} \mathbb{E}\left[a_i\right]^2 \mathbb{Var}\left[W_{ki}\right] + \mathbb{E}\left[W_{ki}\right]^2 \mathbb{Var}\left[a_i\right] + \mathbb{Var}\left[a_i\right] \mathbb{Var}\left[W_{ki}\right] + \mathbb{Var}\left[b_k\right] \tag{37}$$

### B.3  Derivation for Kronecker Covariance Structure

In Kronecker approximation, the Hessian is represented in Kronecker product form:

$$\boldsymbol{h} = \boldsymbol{A} \otimes \boldsymbol{B} \tag{38}$$

Denote the prior precision as $\lambda^2$, then the posterior precision is

$$\mathbf{P} = \boldsymbol{h} + \lambda^2 \mathbf{I} = \boldsymbol{A} \otimes \boldsymbol{B} + \lambda^2 \mathbf{I} \tag{39}$$

To improve numerical stability, an eigen-decomposition is often performed on $\boldsymbol{A}$ and $\boldsymbol{B}$ in `Laplace Redux` library:

$$\mathbf{P} = (\boldsymbol{U}_A \boldsymbol{\Lambda}_A \boldsymbol{U}_A^\top) \otimes (\boldsymbol{U}_B \boldsymbol{\Lambda}_B \boldsymbol{U}_B^\top) + \lambda^2 \mathbf{I} \tag{Definition}$$

$$= (\boldsymbol{U}_A \otimes \boldsymbol{U}_B)(\boldsymbol{\Lambda}_A \otimes \boldsymbol{\Lambda}_B)(\boldsymbol{U}_A \otimes \boldsymbol{U}_B)^\top + \lambda^2 \mathbf{I} \qquad ((\mathbf{A} \otimes \mathbf{B})(\mathbf{C} \otimes \mathbf{D}) = (\mathbf{AC}) \otimes (\mathbf{BD}))$$

For computational efficiency, for our forward pass we will represent the covariance as $\boldsymbol{C} \otimes \boldsymbol{D}$ form, which results in an approximation:

$$\mathbf{P} \approx (\boldsymbol{U}_A \otimes \boldsymbol{U}_B)((\boldsymbol{\Lambda}_A + \lambda \mathbf{I}_A) \otimes (\boldsymbol{\Lambda}_B + \lambda \mathbf{I}_B))(\boldsymbol{U}_A \otimes \boldsymbol{U}_B)\top \tag{40}$$

$$= \left( \left[ (\boldsymbol{U}_A (\boldsymbol{\Lambda}_A + \lambda \mathbf{I}_A)) \otimes (\boldsymbol{U}_B (\boldsymbol{\Lambda}_B + \lambda \mathbf{I}_B)) \right] (\boldsymbol{U}_A \otimes \boldsymbol{U}_B)^\top \right)^{-1} \tag{41}$$

$$= (\boldsymbol{U}_A (\boldsymbol{\Lambda}_A + \lambda \mathbf{I}_A) \boldsymbol{U}_A^\top)^{-1} \otimes (\boldsymbol{U}_B (\boldsymbol{\Lambda}_B + \lambda \mathbf{I}_B) \boldsymbol{U}_B^\top)^{-1} \tag{42}$$

$$= (\boldsymbol{U}_A \otimes \boldsymbol{U}_B)(\boldsymbol{\Lambda}_A \otimes \boldsymbol{\Lambda}_B + \lambda^2 \mathbf{I})(\boldsymbol{U}_A \otimes \boldsymbol{U}_B)^\top + {\color{blue}\lambda \mathbf{I}_A \otimes \boldsymbol{\Lambda}_B + \boldsymbol{\Lambda}_A \otimes \lambda \mathbf{I}_B}, \tag{43}$$

where the extra term introduced by the approximation is written in blue colour.

Recall for an efficient implementation for computing $\sum_{1 \leq i,j \leq \mathrm{D_{in}}} \mathbb{Cov}[a_i W_{ki}, a_j W_{lj}]$ in Eq. (30), we need to retrieve the covariance between the $k^{\text{th}}$ row of weight and $l^{\text{th}}$ row of weight, which is a $\mathrm{D_{in}} \times \mathrm{D_{in}}$ matrix:

$$\mathbb{Cov}\left[\boldsymbol{W}[k, :], \boldsymbol{W}[l, :]\right] = \begin{bmatrix} \mathbb{Cov}[W_{k1}, W_{l1}] & \dots & \mathbb{Cov}[W_{k1}, W_{l\,\mathrm{D_{in}}}] \\ \vdots & \vdots & \vdots \\ \mathbb{Cov}[W_{k\,\mathrm{D_{in}}}, W_{l1}] & \dots & \mathbb{Cov}[W_{k\,\mathrm{D_{in}}}, W_{l\,\mathrm{D_{in}}}] \end{bmatrix} \tag{44}$$

However, for posterior stored in Kronecker product form, we will have $D_{\text{in}} \times D_{\text{in}}$ numbers of $D_{\text{out}} \times D_{\text{out}}$ matrix, which complicates the retrieval of $\mathbb{C}\text{ov}[\boldsymbol{W}[k,:], \boldsymbol{W}[l,:]]$.

## B.4 Derivation for Activation Layers

For $\boldsymbol{a} = g(\boldsymbol{h})$ where $\boldsymbol{h} \sim \mathcal{N}(\boldsymbol{h}; \mathbb{E}[\boldsymbol{h}], \boldsymbol{\Sigma}_h)$ and $g(\cdot)$ is the activation function, we use local linearisation to approximate the distribution of $\boldsymbol{a}$. Specifically, we do a first-order Taylor expansion on $g(\cdot)$ at $\mathbb{E}[\boldsymbol{h}]$:

$$\boldsymbol{a} = g(\boldsymbol{h}) \tag{45}$$

$$\approx g(\mathbb{E}[\boldsymbol{h}]) + \boldsymbol{J}_g|_{\boldsymbol{h}=\mathbb{E}[\boldsymbol{h}]}(\boldsymbol{h} - \mathbb{E}[\boldsymbol{h}]). \tag{46}$$

Given that Gaussian distribution is closed under linear transformation, we have

$$\boldsymbol{h} \sim \mathcal{N}(\mathbb{E}[\boldsymbol{h}], \boldsymbol{\Sigma}_h) \tag{47}$$

$$\boldsymbol{h} - \mathbb{E}[\boldsymbol{h}] \sim \mathcal{N}(\boldsymbol{0}, \boldsymbol{\Sigma}_h) \tag{48}$$

$$\boldsymbol{J}_g|_{\boldsymbol{h}=\mathbb{E}[\boldsymbol{h}]}(\boldsymbol{h} - \mathbb{E}[\boldsymbol{h}]) \sim \mathcal{N}(\boldsymbol{0}, \boldsymbol{J}_g|_{\boldsymbol{h}=\mathbb{E}[\boldsymbol{h}]}^\top \boldsymbol{\Sigma}_h \boldsymbol{J}_g|_{\boldsymbol{h}=\mathbb{E}[\boldsymbol{h}]}) \tag{49}$$

$$g(\mathbb{E}[\boldsymbol{h}]) + \boldsymbol{J}_g|_{\boldsymbol{h}=\mathbb{E}[\boldsymbol{h}]}(\boldsymbol{h} - \mathbb{E}[\boldsymbol{h}]) \sim \mathcal{N}(g(\mathbb{E}[\boldsymbol{h}]), \boldsymbol{J}_g|_{\boldsymbol{h}=\mathbb{E}[\boldsymbol{h}]}^\top \boldsymbol{\Sigma}_h \boldsymbol{J}_g|_{\boldsymbol{h}=\mathbb{E}[\boldsymbol{h}]}) \tag{50}$$

$$\boldsymbol{a} \underset{\text{approx}}{\sim} \mathcal{N}(\boldsymbol{a}; g(\mathbb{E}[\boldsymbol{h}]), \boldsymbol{J}_g|_{\boldsymbol{h}=\mathbb{E}[\boldsymbol{h}]}^\top \boldsymbol{\Sigma}_h \boldsymbol{J}_g|_{\boldsymbol{h}=\mathbb{E}[\boldsymbol{h}]}). \tag{51}$$

## B.5 Probit Approximation for Classification

Following [4], in classification we treat the logits before last layer activation (softmax) as model output $f$. Then we can use probit approximation to get posterior predictive:

Binary [14, 25]

$$p(y^* \mid x^*) = \int_{\mathbb{R}} \text{sigmoid}(f^*) \mathcal{N}\left(f^* \mid \mu^*, \sigma^{*2}\right) \mathrm{d}f^* \tag{52}$$

$$\approx \int \Phi(f^*) \mathcal{N}\left(f^* \mid \mu^*, \sigma^{*2}\right) \mathrm{d}f^* \tag{53}$$

$$= \sigma\left(\frac{\mu^*}{\sqrt{1 + \frac{\pi}{8}\sigma^{*2}}}\right). \tag{54}$$

Multi-class [7]

$$p(\mathbf{y}^* \mid \mathbf{x}^*) = \int_{\mathbb{R}^C} \text{softmax}(\mathbf{f}^*) \mathcal{N}(\mathbf{f}^* \mid \mu^*, \boldsymbol{\Sigma}^*) \mathrm{d}\mathbf{f}^*$$

$$\overset{\text{j-th element}}{\approx} \frac{\exp(\tau_i)}{\sum_{j=1}^{C} \exp(\tau_j)}, \text{ where } \tau_j = \frac{\mu_j^*}{\sqrt{1 + \frac{\pi}{8}\Sigma_{jj}^*}} \tag{55}$$

## B.6 Transformer Block

There are four components in each transformer block [26]: (1) multi-head attention; (2) MLP; (3) layer normalisation; and (4) residual connection. For MLP blocks, the propagation is the same as described above. For layer normalisation, as Gaussian distribution is closed under linear transformation, push distribution over it is straightforward. For residual connection, we assume the input and output are independent. We describe how to push distribution through attention layers below. Note for computational reasons, we always assume the input has diagonal covariance.

**Attention Block** Given an input $\boldsymbol{H} \in \mathbb{R}^{T \times D}$ where $T$ is the number of tokens in the input sequence and $D$ is the dimension of each token, denote the query, key and value matrices as $\boldsymbol{W}_Q \in \mathbb{R}^{D \times D}$, $\boldsymbol{W}_K \in \mathbb{R}^{D \times D}$, $\boldsymbol{W}_V \in \mathbb{R}^{D \times D}$ respectively, the key, query and value in an attention blocks are

$$\boldsymbol{Q} = \boldsymbol{H}\boldsymbol{W}_Q, \quad \boldsymbol{K} = \boldsymbol{H}\boldsymbol{W}_K, \quad \boldsymbol{V} = \boldsymbol{H}\boldsymbol{W}_V, \tag{56}$$

and the output of attention block is

$$\text{Attention}(\boldsymbol{H}) = \text{Softmax}(\frac{\boldsymbol{Q}\boldsymbol{K}^\top}{\sqrt{D}})\boldsymbol{V}. \tag{57}$$

When the input $\boldsymbol{H}$ is a distribution, $\boldsymbol{Q}$, $\boldsymbol{K}$ and $\boldsymbol{V}$ will all be distributions as well. As pushing a distribution over a softmax activation requires further approximation, we ignore the distribution over $\boldsymbol{Q}$ and $\boldsymbol{K}$ for computational reasons and compute their value by using the mean of input:

$$\boldsymbol{Q} = \mathbb{E}\left[\boldsymbol{H}\right]\mathbb{E}\left[\boldsymbol{W}_Q\right], \quad \boldsymbol{K} = \mathbb{E}\left[\boldsymbol{H}\right]\mathbb{E}\left[\boldsymbol{W}_K\right]. \tag{58}$$

For $\boldsymbol{V}$, for simplicity we describe our approximation for a single token $\boldsymbol{h}$ whose value is $\boldsymbol{v} = \boldsymbol{W}_V\boldsymbol{h}$ with $k^\text{th}$ element being $v_k = \sum_{i=1}^D W_{V_{ki}}h_i$. Assuming $\boldsymbol{h}$ is a Gaussian, the covariance between the $k^\text{th}$ and the $l^\text{th}$ value is

$$\mathbb{C}\text{ov}\left[v_k, v_l\right] = \mathbb{C}\text{ov}\left[\sum_{i=1}^D W_{V_{ki}}h_i, \sum_{j=1}^D W_{V_{lj}}h_j\right] \tag{59}$$

$$= \sum_{i=1}^D \sum_{j=1}^D \mathbb{C}\text{ov}\left[W_{V_{ki}}h_i, W_{V_{lj}}h_j\right]. \tag{60}$$

We have

$$\mathbb{C}\text{ov}\left[v_k, v_l\right] = \sum_{i=1}^D \sum_{j=1}^D \mathbb{C}\text{ov}\left[W_{V_{ki}}h_i, W_{V_{lj}}h_j\right] \quad \text{(definition)}$$

$$= \sum_{i=1}^D \sum_{j=1}^D W_{V_{ki}}W_{V_{lj}}\mathbb{C}\text{ov}\left[h_i, h_j\right] \quad (\boldsymbol{W}_V \text{ deterministic})$$

$$\approx \sum_{i=1}^D W_{V_{ki}}W_{V_{li}}\mathbb{V}\text{ar}\left[h_i\right]. \quad \text{(ignore correlation between } \boldsymbol{h} \text{ for computational reason)}$$

$$\mathbb{V}\text{ar}\left[v_k\right] = \sum_{1 \leq i,j \leq D} \mathbb{C}\text{ov}\left[W_{V_{ki}}h_i, W_{V_{kj}}h_j\right] \quad \text{(definition)}$$

$$\approx \sum_{1 \leq i,j \leq D} \left(\mathbb{E}\left[h_i\right]\mathbb{E}\left[h_j\right] + \mathbb{C}\text{ov}\left[h_i, h_j\right]\right)\mathbb{C}\text{ov}\left[W_{ki}, W_{kj}\right] + \mathbb{E}\left[W_{ki}\right]\mathbb{E}\left[W_{kj}\right]\mathbb{C}\text{ov}\left[h_i, h_j\right]$$

$$\text{(assumption A2)}$$

$$= \sum_{1 \leq i \leq D} \left(\mathbb{E}\left[h_i\right]^2 + \mathbb{V}\text{ar}\left[h_i\right]\right)\mathbb{V}\text{ar}\left[W_{ki}\right] + \mathbb{E}\left[W_{ki}\right]^2\mathbb{V}\text{ar}\left[h_i\right].$$

$$(\boldsymbol{W}_V \text{ is isotropic Gaussian})$$

$$\tag{61}$$

Once we have the distribution over $\boldsymbol{V}$, the distribution over $\text{Attention}(\boldsymbol{H})$ becomes a distribution of linear combination of Gaussian, which is tractable.

Then for multi-head attention, we assume each attention head's output is independent, which allows us to compute the distribution over the final output in tractable form. As we assume all input is isotropic, here we only need to compute the variance for each dimension.

## C   Experiment

### C.1   Regression

Table 4 gives the UCI regression data set information and the neural network structure we used. For all neural networks, we use ReLU activation function. In Table 5 we report the Root Mean Square

Table 4: UCI regression experiment setup.

| Dataset Name | Shorthand | $(n, d)$ | Network Structure |
|---|---|---|---|
| SERVO | SERVO | (167, 4) | $d$-50-1 |
| LIVER DISORDERS | LD | (345, 5) | $d$-50-1 |
| AUTO MPG | AM | (398, 7) | $d$-50-1 |
| REAL ESTATE VALUATION | REV | (414,6) | $d$-50-1 |
| FOREST FIRES | FF | (517, 12) | $d$-50-1 |
| INFRARED THERMOGRAPHY TEMPERATURE | ITT | (1020, 33) | $d$-100-1 |
| CONCRETE COMPRESSIVE STRENGTH | CCS | (1030, 8) | $d$-100-1 |
| AIRFOIL SELF-NOISE | ASN | (1503, 5) | $d$-100-1 |
| COMMUNITIES AND CRIME | CAC | (1994, 127) | $d$-100-1 |
| PARKINSONS TELEMONITORING | PT | (5875, 19) | $d$-50-50-1 |
| COMBINED CYCLE POWER PLANT | CCPP | (9568, 4) | $d$-50-50-1 |

Error (RMSE), Ours results in matching or better performance compared with sampling and GLM, indicating the effectiveness of our method. Note that as the mean of the posterior prediction of our method is the same as the prediction made by setting the weights of the neural network to be the mean of the posterior, we result in the same prediction as GLM of LA, and hence the same performance.

Table 5: Root Mean Square Error ↓ on UCI regression data sets. Ours results in better or matching performance compared with sampling and GLM, indicating the effectiveness of our method.

| | | MFVI (Diag. Cov.) | | Laplace Approximation (Full Cov.) | | |
|---|---|---|---|---|---|---|
| | $(n, d)$ | Sampling | Ours | Sampling | GLM | Ours |
| SERVO | (167, 4) | $0.749_{\pm0.147}$ | $\mathbf{0.740}_{\pm0.143}$ | $1.632_{\pm0.233}$ | $\mathbf{0.658}_{\pm0.141}$ | $\mathbf{0.658}_{\pm0.141}$ |
| LD | (345, 5) | $\mathbf{0.884}_{\pm0.273}$ | $\mathbf{0.881}_{\pm0.272}$ | $0.989_{\pm0.441}$ | $\mathbf{0.977}_{\pm0.418}$ | $\mathbf{0.977}_{\pm0.418}$ |
| AM | (398, 7) | $\mathbf{0.415}_{\pm0.115}$ | $\mathbf{0.417}_{\pm0.113}$ | $0.505_{\pm0.105}$ | $\mathbf{0.371}_{\pm0.103}$ | $\mathbf{0.371}_{\pm0.103}$ |
| REV | (414, 6) | $\mathbf{0.563}_{\pm0.096}$ | $\mathbf{0.562}_{\pm0.095}$ | $0.789_{\pm0.130}$ | $\mathbf{0.532}_{\pm0.104}$ | $\mathbf{0.532}_{\pm0.104}$ |
| FF | (517, 12) | $\mathbf{0.874}_{\pm1.123}$ | $\mathbf{0.874}_{\pm1.124}$ | $\mathbf{0.910}_{\pm0.824}$ | $\mathbf{0.852}_{\pm0.792}$ | $\mathbf{0.852}_{\pm0.792}$ |
| ITT | (1020, 33) | $\mathbf{0.481}_{\pm0.057}$ | $\mathbf{0.497}_{\pm0.066}$ | $0.560_{\pm0.075}$ | $\mathbf{0.507}_{\pm0.072}$ | $\mathbf{0.507}_{\pm0.072}$ |
| CCS | (1030, 8) | $\mathbf{0.472}_{\pm0.102}$ | $\mathbf{0.476}_{\pm0.106}$ | $0.494_{\pm0.102}$ | $\mathbf{0.301}_{\pm0.057}$ | $\mathbf{0.301}_{\pm0.057}$ |
| ASN | (1503, 5) | $0.568_{\pm0.062}$ | $\mathbf{0.560}_{\pm0.062}$ | $0.550_{\pm0.069}$ | $\mathbf{0.352}_{\pm0.055}$ | $\mathbf{0.352}_{\pm0.055}$ |
| CAC | (1994, 127) | $\mathbf{0.571}_{\pm0.105}$ | $0.585_{\pm0.092}$ | $1.481_{\pm0.167}$ | $\mathbf{0.703}_{\pm0.101}$ | $\mathbf{0.703}_{\pm0.101}$ |
| PT | (5875, 19) | $0.601_{\pm0.067}$ | $\mathbf{0.590}_{\pm0.068}$ | $0.479_{\pm0.081}$ | $\mathbf{0.410}_{\pm0.076}$ | $\mathbf{0.410}_{\pm0.076}$ |
| CCPP | (9568, 4) | $\mathbf{0.241}_{\pm0.038}$ | $\mathbf{0.241}_{\pm0.038}$ | $0.358_{\pm0.041}$ | $\mathbf{0.224}_{\pm0.037}$ | $\mathbf{0.224}_{\pm0.037}$ |
| Bold Count | | 8/11 | 10/11 | 2/11 | 11/11 | 11/11 |

## C.2 Classification

Table 6 gives the classification data sets information and the neural network structure we used. We use ReLU activation for MLP. For ViT, we make the MLP block in the last two transformer block and the classification head Bayesian, and treat the rest of the weight deterministically. In Table 7 we report the test accuracy, our method results in matching or better performance compared with sampling and GLM, indicating the effectiveness of our method.

Table 6: Classification experiment setup.

| Dataset Name | $(n, d)$ | Network Structure |
|---|---|---|
| MNIST | (50000, 784) | $d$-128-64-10 |
| FMNIST | (50000, 784) | $d$-128-64-10 |
| CIFAR-10 | (50000, 3, 32, 32) | ViT-base |
| CIFAR-100 | (50000, 3, 32, 32) | ViT-base |

In Fig. 4, we show kernel density plots over the predictive entropy of an FMNIST-trained MLP evaluated on MNIST. Our method can distinguish between in-distribution and OOD data better than the LA MAP and MFVI Sampling. Although our method underfits on the in-distribution data, the separation between is clear for the OOD data similar.

Table 7: Accuracy ↑ on classification data sets. Ours results in better or matching performance compared with sampling and GLM, indicating the effectiveness of our method.

| | | MFVI (Diag. Cov.) | | LA (Kron. Cov. for MLP, Diag. Cov. for ViT) | | |
| | | Sampling | Ours | Sampling | GLM | Ours |
|---|---|---|---|---|---|---|
| MNIST | MLP | $\mathbf{0.974}_{\pm 0.002}$ | $\mathbf{0.974}_{\pm 0.002}$ | $0.972_{\pm 0.002}$ | $\mathbf{0.975}_{\pm 0.002}$ | $\mathbf{0.975}_{\pm 0.002}$ |
| FMNIST | MLP | $\mathbf{0.843}_{\pm 0.004}$ | $\mathbf{0.842}_{\pm 0.004}$ | $0.868_{\pm 0.004}$ | $\mathbf{0.882}_{\pm 0.003}$ | $\mathbf{0.881}_{\pm 0.003}$ |
| CIFAR-10 | ViT | $\mathbf{0.978}_{\pm 0.001}$ | $\mathbf{0.978}_{\pm 0.001}$ | $0.971_{\pm 0.002}$ | $\mathbf{0.974}_{\pm 0.002}$ | $\mathbf{0.976}_{\pm 0.002}$ |
| CIFAR-100 | ViT | $\mathbf{0.896}_{\pm 0.003}$ | $\mathbf{0.895}_{\pm 0.003}$ | $0.855_{\pm 0.004}$ | $0.873_{\pm 0.003}$ | $\mathbf{0.884}_{\pm 0.003}$ |

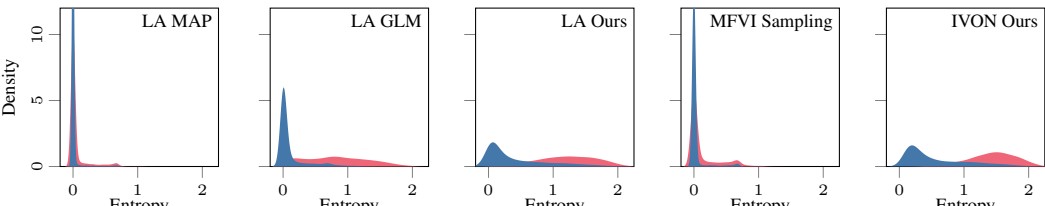

Figure 4: Kernel density plots over the predictive entropy from an MLP trained on FMNIST (blue, in-distribution) and data from MNIST (red, out-of-distribution). Our method results in a clear separation between the in- and out-of-distribution data.

