# OpenReview forum: "Posterior Inferred, Now What? Streamlining Prediction in Bayesian Deep Learning"
_NeurIPS.cc/2024/Workshop/BDU — NeurIPS BDU Workshop 2024 Poster_

### Official Review · Reviewer_rqad · 2024-09-26
**The overall presentation of the paper is good with quite broad experiment results. The paper's contribution is valuable, especially since the topic is still underexplored.**

**Rating:** 8
**Confidence:** 3

**Review:**

The paper proposes a new technique to approximate the predictive distribution in Bayesian Deep Learning (BDL) given an approximate posterior. The introduced technique is faster compared to the standard Monte Carlo (MC) sample-based method. The core idea is to make a Gaussian approximation for the neural network's linear layer and perform local linearizations on the activation function.

The overall presentation of the paper is good with quite broad experiment results. The paper's contribution is valuable, especially since the topic is still underexplored.

some comments:

1. Any comments on why only the last two transformer blocks and the classification head are considered here? For example, what are the trade-offs regarding speed and performance in this context? it might be good to mention this in the paper.
>we make the MLP block in the last two transformer block 282 and the classification head Bayesian, and treat the rest of the weight deterministically.
2. Currently, two types of neural network architectures are considered, both of which are widely used in various applications today. What are the possibilities of using other architectures for this? In other words, how general is the proposed method?
3. In Figure 3, IVON is mentioned, but I couldn’t find any reference to it in the main text. Is the IVON mentioned from Shen et al. (https://arxiv.org/abs/2402.17641)? I think this should be fixed in the camera-ready version.

---

### Official Review · Reviewer_FgVi · 2024-10-06
**An okay submission, but not fully convinced about the need for the method or its performance**

**Rating:** 5
**Confidence:** 3

**Review:**

## Main review

In this paper, the authors propose to propagate the uncertainty that comes from the posterior over the weights of a Bayesian neural network by applying an approximation to the weight-activation products and a local linear approximations to the activation functions.

This approach is interesting, but it is also not fully new (see https://arxiv.org/pdf/1810.03958). The experiments seem reasonably thorough, but the report of error bars (and marking of "winning") methods is unclear. Generally, I am not convinced that the proposed method offers a clear performance advantage over, say GLM, or why one would need this approach in the first place, compared to simply using GLM. A clearer framing of what the proposed approach offers that existing linearisation methods do not is important here.

## Suggested minor adjustments

L2-3: “efficient computation of inferences”: Phrasing here is a little off. Predictions are not typically viewed as something that is inferred.

L7: “layers. Thus allowing…”: probably a comma instead of a full stop.

L14: “identifying failure modes, and identify”: identify repeated.

L19-20: “ function itself). For example…”: writing here reads a bit off as well. Maybe comma instead of full stop?

L20-21: “secondly make inferences of interest based on the estimated posterior”: again, as per one of the previous comments, we typically make inferences about the posterior itself or, in general, latent variables, not downstream quantities (predictions) as seems to be implied here.

L25-26: “remain to be the default”: ungrammatical. Maybe “remain the default”?

L27: “prohibited”: ungrammatical. Maybe “prohibitive”? Also, your wider argument here that “sophisticated sampling methods are usually computationally prohibitive” is not very convincing, and virtually none uses vanilla Monte-Carlo for integrating out the posterior distribution.

Figure 1: The authors claim their method’s predictive uncertainty is somehow “better” but do not support this with an argument. I don’t think this obvious from the figure.

L43: “our method result”: typo, should be “results.”

L46: the notation E[theta] is unclear. What distribution are you taking the expectation with? Why not just use a parameter, e.g. mu, here?

L56: the “assumption” that “aW” is Gaussian isn’t an assumption, it’s an approximation. If a and W are Gaussian, their product is *not* Gaussian. I would rephrase this to say, that “we assume W, b, and a are Gaussian, and that a and W are uncorrelated. While aW is non-Gaussian, we will later approximate its distribution by a Gaussian distribution.” Also, the assumption that a and W are independent is not really an assumption: a and W really are independent (for most standard feedforward architectures), since the activations that are input to a layer do not depend on the weights of the layer.

L58: “Conseuqently”: typo, “Consequently”

Figure 3 (a): legend hides half of the plot, consider rearranging.

Figure 3 (b): histograms overlap each other and make it hard to read the plot. Also the histogram bars are unequal sizes.

Tables 1 and 2: good to see error bars, but what do they mean? Are these 95% confidence intervals on the NLPD? In most cases, your error bars overlap with the error bars of the sampling method, in which case you should probably boldface both methods.

L113: “propgation”: typo, “propagation.”

L115: “obtain”: typo, “obtrains” (also repeating obtain).

---

### Decision · Program_Chairs · 2024-10-09

Accept (Poster)